# Disentangling the roles of different vector species during a malaria resurgence in Eastern Uganda

Max McClure[1]*, Ambrose Oruni[2], Emmanuel Arinaitwe[3], Alex Musiime[4],
Patrick Kyagamba[3], Geoffrey Otto[3], James Adiga[3], Jackson Asiimwe Rwatooro[3],
Maxwell Kilama[3], Paul Krezanoski[1], Jessica Briggs[1], Philip J. Rosenthal[1],
Joaniter I. Nankabirwa[3,5], Moses R. Kamya[3,5], Grant Dorsey[1], Bryan Greenhouse[1],
Isabel Rodriguez-Barraquer[1,6]

1 Department of Medicine, University of California San Francisco, San Francisco, California, United States
of America, 2 Department of Vector Biology, Liverpool School of Tropical Medicine, Liverpool, United
Kingdom, 3 Infectious Diseases Research Collaboration, Kampala, Uganda, 4 National Malaria Control
Division, Ministry of Health, Kampala, Uganda, 5 School of Medicine, Makerere University, Kampala,
Uganda, 6 Chan Zuckerberg Biohub, San Francisco, California, United States of America

* max.mcclure@ucsf.edu

pgph.0004436

Health, UNITED STATES OF AMERICA

**Peer Review History:** PLOS recognizes the
benefits of transparency in the peer review
process; therefore, we enable the publication
of all of the content of peer review and
author responses alongside final, published
articles. The editorial history of this article is
available here: https://doi.org/10.1371/journal.
pgph.0004436

## Abstract

In 2021–23, a resurgence of malaria occurred in the Tororo District of Uganda following a change in formulations used for indoor residual spraying of insecticide (IRS). Prior analyses showed that this increase was temporally associated with the replacement of *Anopheles gambiae* sensu lato by *An. funestus* as the dominant local vector. To investigate this association, we used data from a cohort of 422 children in 94 households from 2021-2023 in Tororo District and neighboring Busia District, where IRS was not implemented. Participants underwent passive and monthly active surveillance for infection with *Plasmodium falciparum* by quantitative PCR. Mosquitoes were collected in each sleeping room in cohort households every 2 weeks using CDC light traps. We assessed the association between spatiotemporally smoothed annualized household entomologic inoculation rates (aEIR) and individual *P. falciparum* infections using shared frailty models. Overall, each doubling of the aEIR was associated with a 29% increase in the hazard of *P. falciparum* (adjusted HR 1.29, 95% CI 1.25-1.33). Considering species-specific aEIRs, this effect was primarily driven by *An. funestus*: a doubling of *An. funestus* aEIR was associated with a 29% increase in hazard rate (1.29, 1.25-1.34), while the association was smaller for *An. gambiae* (1.04, 1.01-1.08). These relationships were stronger in Tororo than in Busia. These results support the inference that the replacement of *An. gambiae* with *An. funestus* was a driver of increased malaria incidence in Tororo District and demonstrates associations between household-level entomological data and risk of *P. falciparum* infection.

**Data availability statement:** The dataset underlying this study is available in the ClinEpiDB database: https://clinepidb.org/ce/app/workspace/analyses/DS_17191d35b9.

**Funding:** Funding was provided by the National Institutes of Health as part of the International Centers of Excellence in Malaria Research (ICEMR) program (U19AI089674 to GD) and the UCSF Biology of Infectious Diseases Training Program (2T32AI007641 to GD). The funders had no role in study design, data collection and analysis, decision to publish, or preparation of the manuscript.

**Competing interests:** The authors have declared that no competing interests exist.

## Introduction

In 2021–23, following a five-year decline, malaria incidence notably resurged in the Tororo District of Uganda, increasing from 0.36 to 2.97 episodes per year in one region. This trend coincided with a transition from organophosphate to clothianidin-based indoor residual spraying (IRS) insecticides in March 2020, accompanied by replacement of *Anopheles gambiae* sensu lato (s.l.) by *An. funestus*, a species found to be relatively tolerant to clothianidin, as the principal local malaria vector. Neighboring Busia District, which never implemented IRS, saw no corresponding change in vector species composition or malaria incidence. Based on these findings, we previously suggested that a rising *An. funestus* population led to the increased malaria burden observed in Tororo [1].

It is well established that the success of vector control measures is dependent on the local vector composition: beyond adulticide selection, the effectiveness of control measures including housing modification [2,3], zooprophylaxis [4], and larval source management [5] have been dependent on species-specific vector ecology. This explanation squares with growing recognition of the epidemiological importance of *An. funestus* in eastern and southern Africa [6], but it is based only on temporal correlation. It has been difficult to causally tie mosquito exposures to clinical outcomes or implicate specific vector species in outbreaks [7], but doing so is critical for assessing the value of targeted vector control.

To further characterize the relationship between vector composition and the recent resurgence of malaria in Tororo, we analyzed data from a detailed cohort study to investigate the association between household-level annualized entomological inoculation rate (aEIR) and time to *P. falciparum* infection, considering aEIR for each relevant malaria vector.

## Methods

### Study participation

The PRISM border cohort study was conducted from August 2020 to September 2023 in Tororo and Busia, both of which are historically high malaria transmission districts in eastern Uganda. The region has historically experienced two seasonal peaks in malaria transmission following the rainy seasons, with precipitation peaking in April and November (based on the 2000–2018 mean) [8].

Data was collected in three geographic zones across the two districts: northern Busia, where IRS has never been implemented; "Tororo, near border," a zone located 0.7-3.5km from the border between Tororo and Busia; and "Tororo, away from border," farther north (5.5-10.8km from the border) in Tororo (Fig 1).

Tororo District was subject to multiple vector control interventions since 2013, with concomitant declines in malaria transmission until the 2020 resurgence. Pyrethroid long-lasting insecticidal nets (LLINs) were distributed in 2013, 2017 and June 2020, and a mix of LLINs treated with deltamethrin (Yorkool) and deltamethrin plus piperonyl butoxide (PermaNet 3.0) in August 2023. IRS was conducted with the carbamate bendiocarb from 2014 to 2015, the organophosphate Actellic (pirimiphos-methyl) from

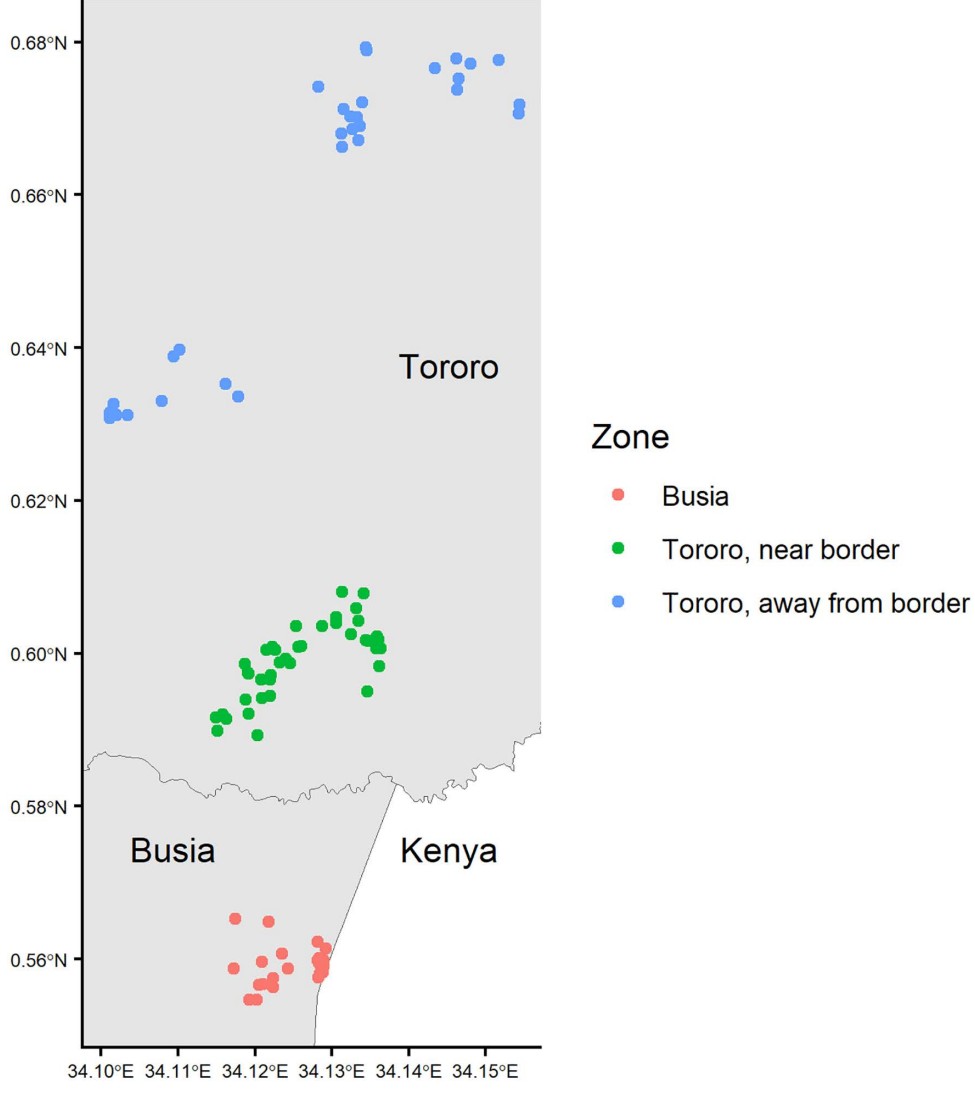

**Fig 1. Household location by zone.**

2017 to 2019, and with Fludora Fusion (clothianidin and deltamethrin) starting in 2020. During the study period, there were three additional rounds of IRS in Tororo: Fludora Fusion in March 2021, SumiShield (clothianidin) in March 2022, and a return to Actellic in March 2023, which was followed by a decline in malaria transmission [1]. In Busia, pyrethroid LLINs were distributed in 2013 and 2017, LLINs treated with deltamethrin plus piperonyl butoxide (PermaNet 3.0) in December 2020, and LLINs treated with alphacypermethrin and clorfenapyr (Interceptor G2) in November 2023, but no IRS was performed (S1 Fig). In the adjoining Kenyan county of Busia, which we did not survey, modeled annual incidence rates from the Malaria Atlas Project suggest that there was no similar rise in malaria incidence during the period covered in this study [9]. The President's Malaria Initiative funded mass distribution of pyrethroid-piperonyl-butoxide ITNs in the county in 2021, but did not implement IRS during the study period [10].

Household and participant enrollment is described in detail elsewhere [1]. Briefly, households containing at least two children ages 5 or younger were randomly selected from within the study areas. Household locations were mapped using

handheld global positioning system (GPS) units. Enrollment was dynamic over the course of the study – children from participating households joined the study when they were born or their households were enrolled.

## Data collection

Routine clinical visits were conducted every 4 weeks. In addition, participants were encouraged to present to the study clinic (open 7 days a week) for evaluation of any illnesses. All routine visits and unscheduled visits when participants were febrile or reported fever in the last 24 hours included collection of thick blood smears and testing of DNA extracted from whole blood for *P. falciparum* DNA using highly sensitive quantitative PCR (qPCR) targeting the var gene acidic terminal sequence, with a limit of detection of 0.03 to 0.15 parasites/μl [11].

Participants with objective or reported fever who tested positive on the thick blood smear were treated for malaria according to national guidelines.

Mosquito collections were conducted from 7pm to 7am every 2 weeks in all study participants' bedrooms using CDC light traps. All mosquitoes underwent morphologic species identification. Up to 50 mosquitoes per CDC light trap collection were assessed for sporozoites using ELISA [12].

## Data analysis

The objective of this analysis was to characterize associations between species-level entomological surveillance data and time from treatment or natural clearance of infection to incident *P. falciparum* infection. First, we developed spatiotemporal models of entomological data to obtain smoothed estimates of species-specific household aEIR, a standard measure of malaria exposure. The smoothing process was intended to minimize the effects of measurement error, zero inflation, and transient changes in entomological measures with little relevance to human exposure. [13] We then used the predictions of these models to assess the relationship between species-specific aEIR and time to infection.

## EIR modeling

Mosquito counts for the region's two major vectors, the *An. gambiae* complex and *An. funestus* subgroup, were modeled as negative binomial temporal generalized additive models (GAMs) with the mgcv package [14] in R Statistical Software (v4.4.1; R Core Team 2024), with date of collection as a smoothed predictor using thin plate splines and including household as a random effect.

Sporozoite rates within the subset of each morphologic species that underwent ELISA were modeled as temporal binomial GAMs, again using thin plate splines and including household as a random effect. For both vector counts and sporozoite rates, separate models were formulated for each geographic zone (i.e., three models) due to their distance and differing vector control intervention histories. No spatial component was included within zones due to the small number of households and the relatively small distances between households within each site; models including household random effects performed better than those with an independent spatial smooth by Akaike information criterion (AIC) for all zones and models (Tables A-D in S1 Text). Additional model types were considered as detailed in the supplementary materials and compared by Akaike information criterion (AIC). Residuals for each model were assessed for deviation from the expected distribution with the DHARMa package [15] in R. After model fitting, the estimated EIR for each species was calculated as the product of the modeled daily vector count and sporozoite rate and then annualized.

## Survival analysis

To investigate the association between entomological metrics and time to incident *P. falciparum* infection, we fit shared frailty models clustered at the level of the individual. Incident *P. falciparum* infection was defined as the first occurrence of either positive qPCR or positive microscopy, regardless of the presence of symptoms, following either a prior documented

malaria treatment course or a spontaneous clearance (i.e., asymptomatic individuals who clear parasitemia without treatment). Spontaneous clearances were defined as three negative qPCR results in succession (covering approximately 3 months, based on sampling every 28 days as previously prescribed [16]). An alternate, more conservative analysis in which we only considered incident infections occurring after documented treatment (intended to address the possibility that persistent low-density parasitemia temporarily falling below the threshold of detection by qPCR could be falsely labeled as a clearance and subsequent reinfection) was also conducted and is reported as a supplementary result. Instances of low parasite density infections detected only by qPCR within 30 days of a prior treated infection were not considered incident infections as these could reflect residual gametocytemia following treatment.

Recurrent events were handled with a gap time approach, assuming independence between all observed event times: day 0 for each period of interest was defined as day 15 following the patient's most recent malaria treatment or day 1 after natural clearance. To deal with interval-censoring of the outcome times, the time of infection was imputed to the midpoint between the most recent negative clinical evaluation and the day of diagnosis.

To minimize the impact of variations in host susceptibility on infection risk, we limited this analysis to children up to 15 years of age. Participants who were positive at all visits, with no documented treatment for malaria or spontaneous clearance, as well as participants followed for fewer than three months were excluded from the analyses.

Models were fit using the frailtyEM package [17] in R. We conducted analyses both for all sites combined and separately by district (i.e., two models, combining the two Tororo zones due to their shared vector control intervention histories). Daily $log_2$-transformed species-specific modeled aEIRs were included as time-dependent covariates, lagged by 14 days to account for the *P. falciparum* intrinsic incubation period; 28-day lags for entomological covariates were also tested and are reported as supplementary results. Models included fixed effects for each of three previously identified time periods representing distinct epidemiological moments: before, during and after malaria resurgence (September 2020-August 2021, September 2021-March 2023, and April-September 2023, respectively) [1], as well as age. We also fit models that considered interactions between aEIRs and the epidemiological time periods. Reported incidence rate ratios are adjusted for all listed covariates unless otherwise specified.

### Ethics statement

Ethical approval was obtained from the Makerere University School of Medicine Research and Ethics Committee (REF 2019–134), the Uganda National Council for Science and Technology (HS 2700), and the University of California, San Francisco Committee on Human Research (257790). Written informed consent was obtained for all participants prior to enrollment into the cohort study. Written informed consent was obtained from the parent/guardian of each participant under 18 years of age. Participants were recruited from 10-08-2020 to 21-02-2023.

## Results

### Study population and entomological data

The study enrolled 472 participants under age 15 from 94 households, of which 45 were excluded from analysis because of an absence of treatment episodes or natural clearances, and an additional 5 were excluded due to less than 90 days of follow-up. The final dataset from this analysis included 422 participants from 94 households, 49% of whom were male (Table 1). The median age at enrollment was 4.6 years (IQR 2.5-9.0). Participants were followed for a median of 977 days (IQR 565–1125).

### Changes in entomological metrics over the study period

In order to quantify entomological trends associated with the resurgence of malaria in Tororo following the switch in IRS from Actellic to Fludora Fusion and SumiShield in 2022–2023, we compared district-wide household-level

**Table 1. Participant characteristics.**

|  | Busia | Tororo, near | Tororo, away | Overall |
|---|---|---|---|---|
| No. individuals | 111 | 167 | 144 | 422 |
| No. households | 23 | 40 | 31 | 94 |
| % Male | 49.5 | 52.1 | 45.8 | 49.3 |
| Median age in years (IQR) | 4.8 (2.4, 8.9) | 4.6 (2.5, 8.7) | 4.4 (2.2, 9.7) | 4.6 (2.5, 9) |
| Median days followed (IQR) | 787 (561, 1139) | 1124 (438, 1126) | 977 (614, 979) | 977 (565, 1125) |
| Median infections per person-year (IQR) | 4.6 (2.5, 8.2) | 5.7 (3.3, 9.5) | 4.9 (3.4, 7.7) | 5.1 (3, 8.6) |
| Median cases malaria per person-year (IQR) | 1.4 (0.7, 2.8) | 2.1 (1, 3.3) | 2 (1.2, 3.4) | 2 (1, 3.3) |

For the purposes of calculating infection incidence, person-time here omits all days between an incident infection and day 14 following the individual's next malaria treatment (see Methods).

collections from before, during and after the resurgence by dividing collections into previously defined time periods (Table 2; see Methods). In Tororo near the border, median *An. funestus* aEIR rose from a baseline of 0 (IQR 0-7.3) to 8.9 (0-20.8) during the resurgence and returned to 0 (0–0) afterwards. Median nightly *An. funestus* counts remained stable during the resurgence (1.4 [0.9-2.3] before to 1.2 [0.6-2.2] during), while SR rose (0 [0–1] before to 1.7 [0-3.1] during). *An. funestus* aEIR. *An. gambiae* aEIR fell from 14 (0-27.5) to 7.8 (0-11.1) to 0 (0–0) over the same time periods.

In Tororo away from the border, median *An. funestus* aEIR rose from 0 (0–0) to 4.5 (0-11.2) during the resurgence and returned to 0 (0–0) afterwards. Median nightly *An. funestus* counts rose during the resurgence (0.1 [0.1-0.2] before to 0.6 [0.2-1.1] during), as did SR (0 [0–0] before to 1.5 [0-2.5] during).

*An. gambiae* aEIR remained similar over the three time periods (0 [0-5.7]; 0 [0-4.2]; 0 [0–0], respectively). In Busia, which did not experience a resurgence, *An. gambiae* remained dominant over the study period, with a median aEIR of 16.4 (5.2-40.5), compared to an *An. funestus* median aEIR of 6 (0-13.8).

Entomological model comparisons are summarized in Tables A-D in S1 Text. Predictions generated from the best vector density and sporozoite rate models are overlaid on Fig 2a–f. Consistent with descriptive analyses, this model suggests that relative contributions of *An. funestus* to vector composition increased during the resurgence in both Tororo zones, and sporozoite rates generally remained equal or higher in *An. funestus* when compared to those in *An. gambiae* throughout the study period in Tororo.

## Changes in risk of infection over the study period

Median incidence of *P. falciparum* infection was highest in Tororo near the border at 5.7 infections per person-year (IQR 3.3-9.5), followed by Tororo away from the border at 4.9 (3.4-7.7) and Busia at 4.6 (2.5-8.2). Consistent with this result, the median time to incident infection was longer in Busia (29 days [95% CI 22–42]) than in the combined Tororo zones (19 days [18–21]) (S2 Fig). Between the start of enrollment at Tororo away from the border and start of the malaria resurgence as defined above (January-August 2021), the percentage of individuals who were positive for *P. falciparum* by PCR at the first evaluation falling within the analysis period was 42.1% (32/76) in Busia, 53.9% (69/128) in Tororo near the border, and 12.8% (15/117) in Tororo away from the border.

As with entomological measures, infection incidence increased during the resurgence in both Tororo zones. In Tororo near the border, incidence increased from a median of 3.1 (IQR 1.1-5.8) infections per person-year to 11.8 (5.3-18.7) and then declined to 7 (2.8-20.3). Away from the border, incidence increased from a median of 1.6 (0.0-1.9) infections per person-year to 9.8 (6.2-15.6) and then declined to 2.5 (0-6.6). Combining both Tororo zones, which experienced the same IRS regimes, incidence increased from 1.9 (0.0-4.1) infections per person-year to 10.8 (5.7-17.8) before declining to 4.0

Table 2. Household-level medians (IQR) for biweekly entomological measures.

| | Busia | | | | Tororo, near | | | | Tororo, away | | | |
|---|---|---|---|---|---|---|---|---|---|---|---|---|
| | Before | During | After | Overall | Before | During | After | Overall | Before | During | After | Overall |
| An. gambiae count | 8.4 (4.6,20.8) | 1.6 (0.8,4.1) | 2.5 (1.3,4.3) | 2.5 (2,7.3) | 7.2 (3.8,11.8) | 0.9 (0.4,1.8) | 0.9 (0.5,2) | 3.5 (1.9,9.1) | 1 (0.6,2.3) | 0.4 (0.2,0.8) | 0.3 (0.1,0.5) | 0.6 (0.3,1) |
| An. funestus count | 0.9 (0.4,1.8) | 0.4 (0.2,0.9) | 1.4 (0.6,3.3) | 0.8 (0.3,1.8) | 1.4 (0.9,2.3) | 1.2 (0.6,2.2) | 0.8 (0.4,2.2) | 1.4 (0.8,2.7) | 0.1 (0.1,0.2) | 0.6 (0.2,1.1) | 0 (0,0.1) | 0.4 (0.1,0.7) |
| An. gambiae SR | 0.8 (0.5,1.1) | 1.3 (0,2.3) | 0 (0,1.9) | 1.2 (0.7,1.8) | 0.5 (0,1) | 1.5 (0,2.9) | 0 (0,0) | 0.7 (0.4,1.2) | 0 (0,0.4) | 0 (0,1.6) | 0 (0,0) | 0 (0,1.8) |
| An. funestus SR | 1.3 (0,2.8) | 0 (0,2.9) | 0 (0,2.5) | 1.8 (0,2.7) | 0 (0,1) | 1.7 (0,3.1) | 0 (0,0) | 0.7 (0,2) | 0 (0,0) | 1.5 (0,2.5) | 0 (0,0) | 1.8 (0,2.6) |
| An. gambiae aEIR | 33.5 (13.2,64.4) | 7.9 (0,13) | 0 (0,32.4) | 16.4 (5.2,40.5) | 14 (0,27.5) | 7.8 (0,11.1) | 0 (0,0) | 9.3 (3.7,19.8) | 0 (0,5.7) | 0 (0,4.2) | 0 (0,0) | 0 (0,2.7) |
| An. funestus aEIR | 8.1 (0,14.5) | 0 (0,7.7) | 0 (0,28.1) | 6 (0,13.8) | 0 (0,7.3) | 8.9 (0,20.8) | 0 (0,0) | 4.6 (0,12.3) | 0 (0,0) | 4.5 (0,11.2) | 0 (0,0) | 2.6 (0,7.1) |
| No. collections | 34 (26,49) | 78 (41,82) | 26 (13,26) | 122 (77,154) | 50 (26,51) | 82 (54,84) | 26 (13,26) | 142 (64,160) | 32 (24,32) | 82 (58,84) | 26 (25,26) | 140 (101,142) |

For the two Tororo collection zones, measures are divided into periods of time "before," "during," and "after" the malaria resurgence described in the text. Species-specific SR reported as percentage. SR = sporozoite rate; aEIR = annualized entomological inoculation rate; No. collections = number of household-nights of mosquito collection.

(0.0-12.0). 26.3% (163/620) of incident infections in our analysis were accompanied by fever in Busia, 28.8% (290/1006) in Tororo near the border, and 38.8% (357/921) in Tororo away from the border.

## Association between entomological metrics and time to incident infection

We fit time to event models to assess the association between entomological metrics and time to incident infection. In survival models adjusting for epidemiologic time period (before, during and after the malaria resurgence), we found a positive association between overall aEIR and *P. falciparum* infection when combining both districts, with a 29% increase in the hazard of infection for each doubling of aEIR (mean adjusted hazard ratio [aHR] 1.29 [95% CI 1.25-1.33], Table 3). When considering species-specific aEIRs, we found that most of this effect was driven by *An. funestus*: a doubling of *An. funestus* aEIR was also associated with a 29% increase in infection rate (1.29 [1.25-1.34]), while the association was smaller for *An. gambiae* (1.04 [1.01-1.08]). Fitting models to districts separately, we observed a stronger relationship between *An. funestus* and infection rate across both Tororo zones (1.27 [1.22-1.32]) than in Busia (1.17 [1.07-1.28]). *An. gambiae* was significantly associated with infection risk when districts were considered individually, though to a lesser extent than *An. funestus* (Tororo: 1.07 [1.02-1.12]; Busia: 1.11 [1.04-1.18]).

When we considered interactions between time periods, the findings for *An. funestus* were present over the entire study. In Tororo, *An. funestus* was significantly associated with infection risk before the resurgence (aHR 1.14), and the association was even stronger during the resurgence (1.30), while the association between infection risk and *An. gambiae* decreased (1.12 before; 1.02 during) (S2 Table). The findings for *An. funestus* were robust when using restricted cubic splines instead of assuming a linear relationship between species-specific aEIR and log hazard. Results were also qualitatively similar when excluding natural clearances, using 28-day-lagged aEIR, or using household SR means rather than modeled SR, with the exception that in these analyses *An. funestus* was not significantly associated with infection risk in Busia (S3 Fig, S1 Table, S3–S5 Tables). In both districts, the fully parameterized model performed better by AIC than a corresponding model without *An. funestus* or *An. gambiae* aEIRs as covariates.

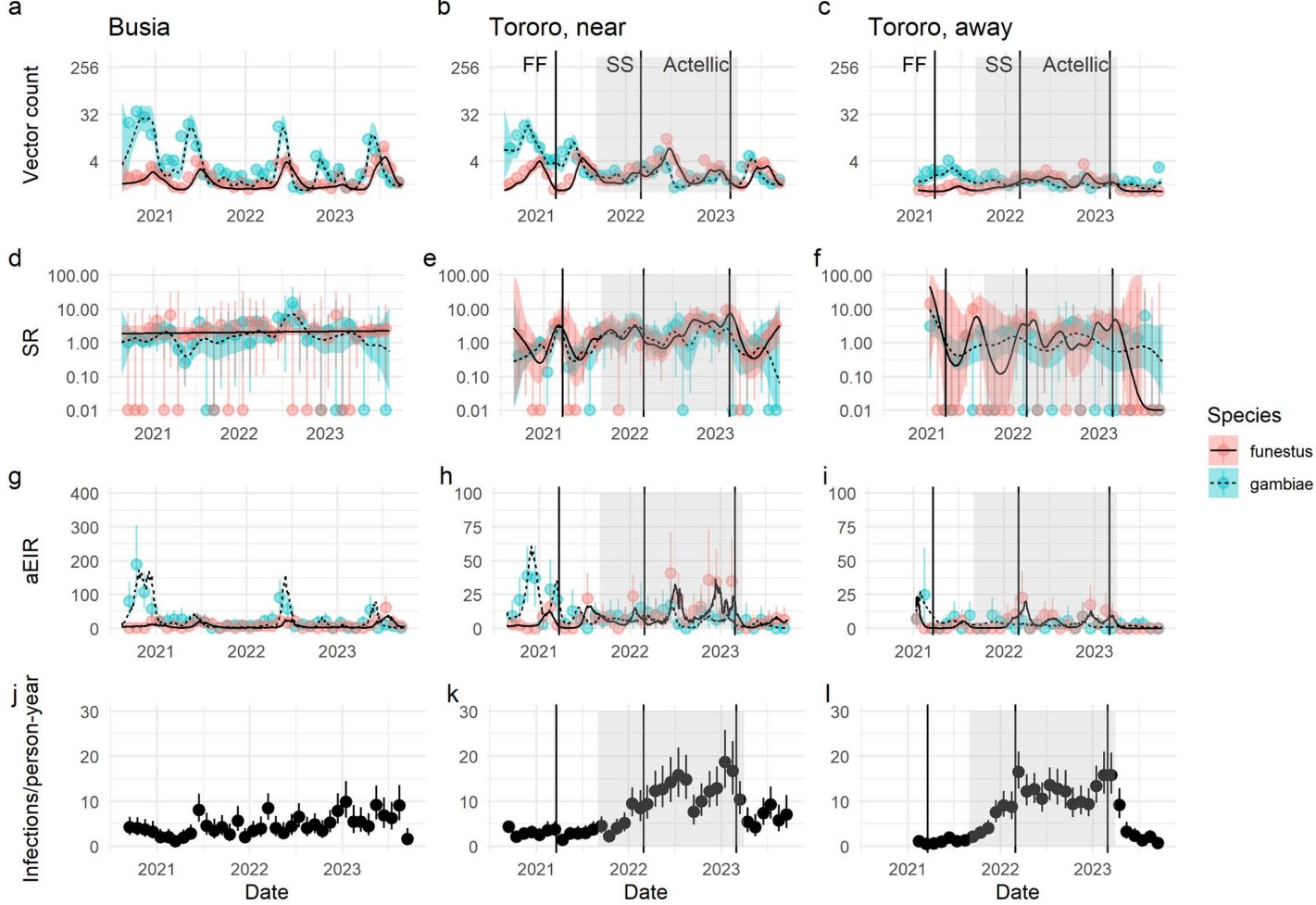

**Fig 2. Temporal trends in entomological measures and infection incidence.** Species-specific vector counts **(a-c)**, species-specific sporozoite rates expressed as a percentage **(d-f)**, species-specific annualized entomological inoculation rate (aEIR) (g-i) and *P. falciparum* infection incidence **(j-l)** by zone (Busia: a, d, g, j; Tororo, near border: b, e, h, k; Tororo, away from border: c, f, i, **l)**. Points with whiskers represent monthly mean crude data and associated uncertainty while lines with confidence bands represent mean model output and associated uncertainty. For vector counts, whiskers show 95% confidence intervals for counts modeled as a Poisson process. For sporozoite rate and malaria incidence, whiskers show 95% confidence intervals from the exact binomial test. For aEIR, whiskers represent 2.5% and 97.5% quantiles of collection-level aEIRs. For GAM output, 95% confidence bands for temporal smooths are shown, excluding household random effects. Note that plots h and i have y axes at a different scale than plot g to emphasize differences in species-specific aEIRs. The timing of indoor residual spraying campaigns is indicated with vertical lines (FF = Fludora Fusion, SS = SumiShield). The period of resurgence described in the main text is indicated with gray shading. For the purposes of calculating infection incidence, person-time here omits all days between an incident infection and day 14 following the individual's next malaria treatment (see Methods).

## Discussion

In Tororo, during a period after a change in IRS insecticide when malaria incidence rose and *An. funestus* became the dominant local malaria vector [1], increased household-level *An. funestus* EIR was a stronger predictor of changes in *P. falciparum* infection rates than *An. gambiae* EIR as measured using CDC light trap collections. This finding supports the inference that the replacement of *An. gambiae* with *An. funestus* was a driver of increased malaria incidence from 2021-23 and demonstrates associations between household-level entomological data and individual infection risk.

**Table 3. aHRs for primary frailty model.**

|  | Busia | Tororo | Overall |
|---|---|---|---|
| Total aEIR |  |  |  |
| Total aEIR | 1.23 (1.17,1.30) | 1.33 (1.270,1.38) | 1.29 (1.25,1.33) |
| Age (years) | 1.05 (1.02,1.09) | 1.02 (0.998,1.03) | 1.02 (1.01,1.04) |
| During | 1.81 (1.43,2.29) | 4.62 (4.010,5.32) | 3.71 (3.30,4.19) |
| After | 1.69 (1.26,2.27) | 2.66 (2.140,3.30) | 2.31 (1.95,2.74) |
| Sp.-specific aEIRs |  |  |  |
| An. funestus aEIR | 1.17 (1.07,1.28) | 1.27 (1.220,1.32) | 1.29 (1.25,1.34) |
| An. gambiae aEIR | 1.11 (1.04,1.18) | 1.07 (1.020,1.12) | 1.04 (1.01,1.08) |
| Age (years) | 1.05 (1.02,1.09) | 1.01 (0.996,1.03) | 1.02 (1.00,1.04) |
| During | 1.70 (1.35,2.15) | 3.72 (3.190,4.33) | 2.97 (2.62,3.37) |
| After | 1.57 (1.13,2.17) | 2.18 (1.760,2.71) | 1.87 (1.57,2.22) |

aHRs assuming a linear relationship between all covariates and the log hazard, using models fit to expected EIR for either Busia, both Tororo zones, or all zones combined. "Total aEIR" denotes models with summed *An. gambiae* and *An. funestus* aEIRs. "Sp.-specific aEIRs" denotes models treating species-specific aEIRs as independent predictors. All EIRs are $\log_2$-transformed.

We have previously provided evidence that the recent dominance of *An. funestus* in Tororo was associated with the development of clothianidin resistance [1,18]. This pattern is analogous to a 2000 KwaZulu-Natal malaria resurgence, which was associated with the invasion of a resistant *An. funestus* population [19]. The present analysis further clarifies the role played by *An. funestus* in driving malaria infections. Increasing household *An. funestus* aEIR was more strongly associated with risk of *P. falciparum* infection both in Tororo, where *An. funestus* populations increased, and in Busia, where they remained stable, and *An. gambiae* s.l. aEIR had smaller associations in both districts.

Our study's strengths lay in the measurement of longitudinal rates of infectious bites and infections, which offered an unusual opportunity to characterize the fundamental relationship between infectious mosquito exposure and *P. falciparum* infection in a real-world setting. We assessed this association at fine spatial and temporal scales, avoiding the imprecision inherent in using proxies for incident infection, such as incident malaria. It is in part because of this precision that we were also able to identify differences in the risk of exposure associated with different vector species.

There were, however, weaknesses inherent to our approach to estimating EIR. First, the relationship between EIR as estimated from field measurements and human exposure is complex. *An. funestus* s.s., the primary malaria vector within the Funestus subgroup, is typically considered endophagic and anthropophilic, meaning overnight indoor CDC light trap captures should be plausible surrogates for its biting behavior [20,21]. In contrast, the *An. gambiae* species complex in eastern Uganda primarily comprises two species with distinct ecologies: *An. gambiae* s.s., another classically endophagic and anthropophilic species, and *An. arabiensis*, a classically exophagic and zoophilic species, though both demonstrate substantial behavioral heterogeneity [6,21,22]. In this study, we were unable to distinguish the two *An. gambiae* sub-species or identify relative changes in proportional representation within the species complex, potentially obscuring important differences in human exposure. The respective roles played by these sub-species may also vary by district: *An. arabiensis* vector densities were greater than equal to those of *An. gambiae* s.s. throughout the study period in both Tororo zones, while the opposite was true in Busia [1]. Our use of an ELISA sporozoite assay that can be less sensitive than molecular assays, inability to quantify mosquito salivary gland sporozoite loads, and focus on *P. falciparum* to the exclusion of other malaria parasites also limited our ability to fully characterize risk associated with exposure to the two vector species.

The effects of vector control measures further complicate these differences between vector species. An additional study from this cohort suggested that, during the period when clothianidin-based IRS formulations were used, the proportion of exposure to mosquitoes occurring outdoors increased for *An. gambiae* s.l. but not for *An. funestus* [23]. A shift towards

outdoor biting by *An. gambiae* s.l. would diminish the utility of indoor captures as surveillance. LLIN use, which may vary between sites, may also have differential effects on the mortality and behavior of different species. *An. funestus* has exhibited both resistance to pyrethroid-treated LLINs [24] and increased day biting in response to LLIN use [25]. If it were present, this behavioral shift could either obscure or amplify an association between estimated EIR and human infection rates, depending on the correlation between the behavior and mosquito captures.

More generally, entomological surveillance at small scales is invariably limited by noise and by the zero inflation that results from estimating sporozoite rates using small sample sizes. Although we attempted to address these issues through smoothing, this difficulty remains a fundamental issue for EIR estimation, particularly with the low nightly mosquito captures we described. Relatively small changes in vector density or sporozoite rates can translate to large changes in estimated EIR, meaning our results should be interpreted with caution. For instance, the changes in *An. funestus* aEIR shown in Table 2 are driven in part by a rise in *An. funestus* SR during the study period. While potential contributors include ecological factors such as increased feeding success or longevity in *An. funestus* or temporarily decreased rates of *An. funestus* emergence with a consequently aging, sporozoite-enriched population, this change could also reflect an artifact of the larger sample size available for sporozoite rate estimation during the resurgence.

It has historically been difficult to correlate household-level measures of a particular malaria vector species, or even anopheline mosquitoes in general, with increased *P. falciparum* infection risk in members of that household. Our results lend further credence to the idea that *An. funestus* was a key driver of the regional malaria resurgence in Tororo, Uganda, while demonstrating associations between fine-scale entomological data with individual risk of *P. falciparum* infection in cohorts with frequent follow-up and longitudinal assessment of parasitemia. This finding supports the role of detailed surveillance in identifying drivers of outbreaks, potentially allowing for targeted control measures.

## Supporting information

**S1 Text. Comparison of candidate entomological generalized additive models by geographic zone (Tables A-D).** X's in columns indicate whether a model contained a spatial smooth or a household random effect. Models are compared using Akaike information criterion (AIC) and percentage of deviance explained. The models selected for further analysis are bolded.
(DOCX)

**S1 Fig. Timeline of interventions.** Timeline of recent indoor residual spraying (IRS) and long-lasting insecticidal net (LLIN) distribution campaigns by district, including names of insecticide formulations and LLIN models.
(PNG)

**S2 Fig. Distribution of times to infection, comparing Busia to combined Tororo zones.** Each boxplot represents the distribution of gap times in days for an individual participant. The boxplot includes marks indicating the median and first and third quartiles, two whiskers, and outlying points, with color varying by site.
(TIFF)

**S3 Fig. Disregarding natural clearances: distribution of times to infection, comparing Busia to combined Tororo zones.** Each boxplot represents the distribution of gap times in days for an individual participant. The boxplot includes marks indicating the median and first and third quartiles, two whiskers, and outlying points, with color varying by site.
(TIFF)

**S1 Table. Disregarding natural clearances: participant characteristics.** For the purposes of calculating infection incidence, person-time here omits all days between an incident infection and day 14 following the individual's next malaria treatment (see Methods).
(DOCX)

**S2 Table. Reanalysis including interactions between time period relative to malaria resurgence and aEIR.** All aEIRs are $\log_2$-transformed.
(DOCX)

**S3 Table. Disregarding natural clearances: aHRs assuming a linear relationship between all covariates and the log hazard.** Models fit to expected EIRs. All aEIRs are $\log_2$-transformed.
(DOCX)

**S4 Table. 28-day lagged aEIR: aHRs assuming a linear relationship between all covariates and the log hazard.** Models fit to expected EIRs. All aEIRs are $\log_2$-transformed.
(DOCX)

**S5 Table. Alternative aEIR: aHRs assuming a linear relationship between all covariates and the log hazard.** aEIR estimated using modeled vector counts and household mean SR. All aEIRs are $\log_2$-transformed.
(DOCX)

## Acknowledgments

We would like to thank the study participants and their families, as well as the study team, Makerere University-UCSF Research Collaboration, and Infectious Diseases Research Collaboration. Portions of this work were performed on the Wynton HPC Co-Op cluster, which is supported by UCSF research faculty and UCSF institutional funds. IRB is a Chan Zuckerberg Biohub Investigator.

## Author contributions

**Conceptualization:** Max McClure, Bryan Greenhouse, Isabel Rodriguez-Barraquer.

**Data curation:** Ambrose Oruni, Alex Musiime, Paul Krezanoski, Jessica Briggs, Grant Dorsey.

**Formal analysis:** Max McClure.

**Funding acquisition:** Philip J. Rosenthal, Joaniter I. Nankabirwa, Moses R. Kamya, Grant Dorsey, Bryan Greenhouse, Isabel Rodriguez-Barraquer.

**Investigation:** Emmanuel Arinaitwe, Patrick Kyagamba, Geoffrey Otto, James Adiga, Jackson Asiimwe Rwatooro, Philip J. Rosenthal, Joaniter I. Nankabirwa, Moses R. Kamya, Grant Dorsey.

**Methodology:** Max McClure, Isabel Rodriguez-Barraquer.

**Project administration:** Emmanuel Arinaitwe, Maxwell Kilama, Joaniter I. Nankabirwa, Moses R. Kamya, Grant Dorsey.

**Supervision:** Emmanuel Arinaitwe, Maxwell Kilama, Paul Krezanoski, Philip J. Rosenthal.

**Writing – original draft:** Max McClure.

**Writing – review & editing:** Max McClure, Ambrose Oruni, Emmanuel Arinaitwe, Alex Musiime, Patrick Kyagamba, Geoffrey Otto, James Adiga, Jackson Asiimwe Rwatooro, Maxwell Kilama, Paul Krezanoski, Jessica Briggs, Philip J. Rosenthal, Joaniter I. Nankabirwa, Moses R. Kamya, Grant Dorsey, Bryan Greenhouse, Isabel Rodriguez-Barraquer.

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
