## [Decision Letter · Decision Letter 0]

4 Jun 2025

PGPH-D-25-00552

Disentangling the roles of different vector species during a malaria resurgence in Eastern Uganda

Dear Dr. McClure,

Thank you for submitting your manuscript to PLOS Global Public Health. After careful consideration, we feel that your manuscript has significant merit but does not fully meet PLOS Global Public Health’s publication criteria as it currently stands. Therefore, we would encourage you to submit a revised version of the manuscript that addresses the points raised during the review process.

Please carefully consider the comments and constructive suggestions by both reviewers especially regarding the methodological aspects and the interpretation of the data given the limitations of the study. 

We look forward to receiving your revised manuscript.

Kind regards,

Amy Kristine Bei

Academic Editor

Journal Requirements:

Reviewers' comments:

Reviewer's Responses to Questions

**Comments to the Author**

1. Does this manuscript meet PLOS Global Public Health’s publication criteria?

Reviewer #1: Yes

Reviewer #2: Yes

2. Has the statistical analysis been performed appropriately and rigorously?

Reviewer #1: Yes

Reviewer #2: Yes

3. Have the authors made all data underlying the findings in their manuscript fully available (please refer to the Data Availability Statement at the start of the manuscript PDF file)?

Reviewer #1: Yes

Reviewer #2: Yes

4. Is the manuscript presented in an intelligible fashion and written in standard English?

Reviewer #1: Yes

Reviewer #2: Yes

Reviewer #1: This is a complicated statistical study of Anopheles vectors captured in different Ugandan districts over the course of malaria resurgence in one district, that the authors are trying to casually link to the change in IRS chemistries from predominately organophosphates, to clothianidin-based, and then back to organophosphates. The foundation of the paper was already laid in the more associative analysis of case and parasitological data at the sites relative to the timeline of the IRS events, vector density and mosquito phenotypic insecticide resistance data published last year (Kamya et al, 2024, PLOS GPH). The paper reads very well and I saw no grammar or confusing text that needed editing. The strengths of the paper are in using a frailty model analysis connecting the time to incident infection of the children studied (adjusted Hazard Ratios) with the annualized EIR data overall, and by species at the two sites, which points to Anopheles funestus as driving the malaria resurgence in Tororo during the implementation of the two clothianidin IRS and then driven back down when OP (Actellic) was used for IRS. It is interesting that these entomological data trends in are not very apparent in simply examining the raw plotted data in Figure 1 compared to the child infection data. I would have liked to know if the entomological data were analyzed by quarter or semi-annually, and if so, were the associations made even stronger? - because there seem to be a couple of peaks each year and maybe different depending on species. The modeling performed seems solid, and the collection of time to incident infection data is particularly strong and analyzed in a couple of different ways to address biases. Overall, the claims made by the paper are backed up by the data presented, and considering the prior published paper. A major weakness of the assumptions is the collection of indoor host-seeking mosquitoes using light trap catches when IRS is the primary control tool and should be acting as a major repellent for those indoor catches in addition to the overall lethal effect, although they appropriately speak to this weakness in the discussion. Another weakness they don’t address that they only tested/account for P. falciparum, and only using an ELISA assay that can be less sensitive than molecular assays. Lastly, the data are lumped together as FF+SS are both clothianidin-based, but infection incidence doesn’t seem to rise until the SS is implemented rather than FF, and the latter contains added pyrethoid and I’m sure the formulations/long-lasting efficacies are different – can this all be more solidly connected to the SS spray alone?

Reviewer #2: Summary

The authors have provided an interesting manuscript detailing further attempts at elucidating the role of different Anopheles vector species in contributing to a malaria resurgence that occurred in Tororo District, Uganda that followed district level changes in insecticide use for Indoor Residual Spraying (IRS) in 2020. Previous work from the same group had suggested this resurgence also coincided with replacement of Anopheles gambiae sensu lato by Anopheles funestus as the primary local vector. Specifically, these same authors have published extensively on this data cohort in this same journal (Kamya et al., 2024) where they initially did all the work to analyze changes in case rates, parasite prevalence, IRS insecticide use, and local vector characteristics between Turoro and Busia Districts.

Here the authors also used data from their prior cohort of 422 children followed from 2021-2023 in two neighboring districts, Turoro (where IRS is implemented) and Busia (no IRS) with passive and monthly active surveillance for Plasmodium falciparum (Pf), and also within the 94 households where these children resided, mosquitoes were collected every 2 weeks using CDC light traps, speciated, and assessed for malaria sporozoite presence in order to calculate annual household entomologic inoculation rates (aEIR). The authors focused in this manuscript on modeling analyses at the household and species level using aEIR and time to infection data. Specifically, they analyzed changes in entomologic metrics (including species-specific aEIR) and changes in infection risk (time to incident infection) over the study period and used this data to model associations, namely the relationship between aEIR and Pf infection risk. They found aEIR increases to be associated with increases in Pf (two-fold increase resulting in 29% HR increase), and that An. funestus aEIR increase was associated with a larger increase in Pf infection HR than for An. gambiae aEIR. Additionally, this An. funestus aEIR association was larger in Tororo than Busia, leading them to conclude this modeling data supports the idea that replacement of An. gambiae with An. funestus drove the malaria resurgence seen in Turoro in 2021-2023. They highlight their unique ability to use longitudinal household-level and species-level EIR data to determine Pf infection risk, which has historically been lacking and is important.

I overall agree with their data presentation, findings, and interpretations. My biggest question has to do with the ability to make significant conclusions based on at times relatively small differences in their model outputs (although the modeling here certainly supports their extensive prior work). In addition, I would ask for some further clarifications (see comments below).

Comments to address

• The manuscript is clear, well-organized, and well written, however there are a few figure legend issues that could be clarified.

o Figure 1 - no legend included? I understand it is a simple figure so may not need it; just want to verify it wasn’t missed.

o Supplemental Figures S1 and S2 – would be helpful to include in the legend the description of the line in the plot. Presumably based on its color it represents the Busia trends of time to infection across individuals. I would want to see the same line for Turoro as well, to better depict comparisons you make in the text (specifically stating time to next infection is longer in Busia).

• Given the proximity of Busia households and the also those in the Turoro border region to Kenya (as evidenced by Figure 1), I think you should also include in the intro or discussion the basics about what is known about malaria cases in this Northwestern Kenya region during the same time frame and their infection control measures.

• Not absolutely necessary, but I would consider the benefit of another component to Figure 1 being a longitudinal depiction of the study timeframe with different IRS insecticide and LLIN distribution interventions highlighted (e.g. arrows above the line for Turoro interventions, arrows below the timeline for Busia interventions)

• Would appreciate better/more explicit description of malaria cases (rates of symptomatic vs asymptomatic) at least in the methods. What percentage of people at baseline enrolment time had asymptomatic parasitemia? You refer to spontaneous clearances a lot without explicitly defining them as presumably untreated asymptomatic people who naturally clear – it would be better to state that (pg 8 line 151-152), as well as to clearly state that 3 consecutive negative screening periods would cover 3 months (based on q28 day sampling?) – which I agree is a reasonable time to assume that newly detected parasitemia is more likely from a new infection. Of course it is possible, as you allude to, that if a person had asymptomatic infection with no treatment required, their subsequent parasitemia or symptomatic case could be from prior parasite versus from new infection from new mosquito bite. I know that you also included data in your supplemental info about a more “conservative” analysis with incident infections only considered after treatment, presumably to account for this scenario. I think it would just overall help to be more clear about in your manuscript about these nuances, but why you would ultimately make the same conclusions – could even just add a sentence in the discussion.

• I also think somewhere in your manuscript you should describe seasonality of malaria cases in this region. Were there any significant weather pattern or case seasonality changes during the resurgence (i.e. changes in rainfall) that should be considered – given An. funestus is slightly more adapted to dryer conditions than An. gambiae?

• Similarly, for more context regarding regional intervention differences – why doesn’t Busia get have IRS implemented? Are there other significant differences in the Busia landscape? What happened with bednet distribution in 2023 (presumably every 3 year planned distribution schedule) – did this occur and could this have affected the subsequent case decline in Turoro again?

• Some clarifications regarding Table 2:

o I understand your data implies there was not a malaria resurgence in Busia, nor changes in vector species, over the period of interest. Still, comparing segmented Turoro data (divided as before/during/after IRS insecticide change) to Busia data collectively over that whole period makes it harder to verify. The Busia data for Table 2 would be helpful to see in a similar before/during/after breakdown (at least in supplemental info) for better comparison.

o It was surprising and somewhat confusing to see aEIR of 0 at so many times – presumably driven by the SR with no infected mosquitoes collected for a given time. Normally would not expect the IQR to be 0 so often given the overall high malaria prevalence in this region? Please clarify your thoughts.

o I would better define the numbers scale in the legend. Presumably number of collections equates to households? What are the An. gambiae or An. funestus counts – an actual number of mosquitoes collected? If so, this is quite low for mosquitoes collected in each house. It is confusing because, looking at the before/during/after of An. funestus count in Turoro near, there is a decline over all times (like An. gambiae, though much less drastic) – which seemingly doesn’t align as well the idea of with An. funestus influx driving a malaria resurgence. This would make it seem the SR changes (increases) for An. funestus are driving aEIR data. It’s more intuitive to think about mosquito numbers changing with insecticide use – but why would An. funestus SR rates change over the study period? This is another reason it would be helpful to see Busia data also divided in time for comparison.

• This previous point brings up the general issue of how the relatively low overall number of sporozoite-infected mosquitoes detected makes it difficult to precisely evaluate temporal trends and make significant conclusions. I appreciate that your modeling work tries to overcome this limitation – but I think it is worth mentioning in your discussion/limitations. (Especially also considering we don’t have any data about quantitative sporozoite burden between An. gambiae and An. funestus here, another factor which can contribute to subsequent infection risk – though I recognize it’s not realistic to assess sporozoite loads with your sample set.)

• Likewise, regarding data in Fig 2 – An. gambiae and An. funestus data look quite similar regarding vector numbers in Turoro areas for much of the main period of interest as do the species-specific SR data especially for Turoro near and Busia. And yet aEIR increases more notably for An. funestus. Hard to see clear pattern in graphed mosquito capture and SR data, but the modeling implies the aEIR changes are bigger. I think this requires more nuanced interpretation of results – making clear that your conclusions are based on aEIR estimates and modeling and noting small differences in vector and SR detection can show up bigger in these EIR-based models, though obviously the infection rise over this period is striking.

o Also, following the data from Figure 2 input to generate the aHR data in Table 3 – it is important to contextualize and note that despite the story and data fitting nicely regarding the switch of IRS insecticide, rise in malaria cases, and noted relative increase in An. funestus mosquito presence, the differences in species-specific aEIR-based aHR calculations between Busia and Turoro are overall still small. Thus, I do especially appreciate the discussion acknowledgement about EIR measures being complex and hard, particularly with An. gambiae having 2 subspecies, and different behavioral characteristics of the various Anopheles vectors and how this could skew the modeling here based on your indoor capture data. It will be interesting to see the data from the publication you mentioned that is under review regarding outdoor exposures. (Nothing requested to change, just a comment.)

• Did you consider using PCR as the method for sporozoite identification rather than ELISA? Would be interesting to have simultaneously looked at presence of different malaria species in your cohort (given some others are finding potential associations between An. funestus and P. ovale, for example (https://www.medrxiv.org/content/10.1101/2024.10.09.24315124v1)).

---

## [Editor Report · Decision Letter 1]

18 Nov 2025

Disentangling the roles of different vector species during a malaria resurgence in Eastern Uganda

PGPH-D-25-00552R1

Dear Dr. McClure,

Thank you for submitting your revised manuscript and for thoughtfully incorporating the suggestions of reviewers. 

We are pleased to inform you that your manuscript 'Disentangling the roles of different vector species during a malaria resurgence in Eastern Uganda' has been provisionally accepted for publication in PLOS Global Public Health!

Best regards,

Amy Kristine Bei

Academic Editor